# REALISMOTION: DECOMPOSED HUMAN MOTION CONTROL AND VIDEO GENERATION IN THE WORLD SPACE

## ABSTRACT

Generating human videos with realistic and controllable motions is a challenging task. While existing methods can generate visually compelling videos, they lack separate control over four key video elements: foreground subject, background video, human trajectory, and action patterns. In this paper, we propose a decomposed human motion control and video generation framework that explicitly decouples motion from appearance, subject from background, and action from trajectory, enabling flexible mix-and-match composition of these elements. Concretely, we first build a ground-aware 3D world coordinate system and perform motion editing directly in the 3D space. Trajectory control is implemented by unprojecting edited 2D trajectories into 3D with focal-length calibration and coordinate transformation, followed by speed alignment and orientation adjustment; actions are supplied by a motion bank or generated via text-to-motion methods. Then, based on modern text-to-video diffusion transformer models, we inject the subject as tokens for full attention, concatenate the background along the channel dimension, and add motion (trajectory and action) control signals by addition. Such a design opens up the possibility for us to generate realistic videos of anyone doing anything anywhere. Extensive experiments on benchmark datasets and real-world cases demonstrate that our method achieves state-of-the-art performance on both element-wise controllability and overall video quality. The source codes and project page are in the supplementary and at `https://anonymous.4open.science/r/RealisMotion-anonymous-3870/`.

## 1 INTRODUCTION

Imagine Mona Lisa participating in a stylish event at a luxurious hotel, gracefully approaching you while holding a glass of red wine. Imagine the real cop Chan shooting the undercover police chief Lau, on a rooftop framed by the Hong Kong skyline. (See Fig. 1 for our results.) While recent advances in human video generation and editing have shown promising results Hu (2024); Zhu et al. (2024); Zhou et al. (2024), existing methods still struggle to realize such creative transformations due to their limited control over individual video elements, such as subject, background, trajectory and action.

Currently, most of the existing human video generation methods are designed to transfer motions between individuals. Given a guidance video and a reference image, these methods first extract motion representations such as pose Yang et al. (2023); Hu (2024) and depth Hu et al. (2025) from the video. Then, they animate the reference image according to the extracted motion. This pipeline, whether operating in 2D image space Wang et al. (2023) or 3D camera space Zhu et al. (2024), is limited in the following aspects. First, the foreground and background are jointly defined, which prevents independent control of the subject and the environment. Second, the tight coupling between action patterns and trajectory prevents independent manipulation of 'what' actions to perform and 'where' to perform them. Third, limited understanding of background geometry hampers editing of the subject's movement along the depth axis, making it hard to produce plausible animations with correct perspective scaling. Fourth, when the camera view changes across frames, the scene coordinate frame also shifts, complicating global trajectory control and consistent action editing. Together, these constraints lead most methods to assume that the human in both guidance video and reference image is centrally framed and near the camera, effectively reducing the task to simple motion copying.

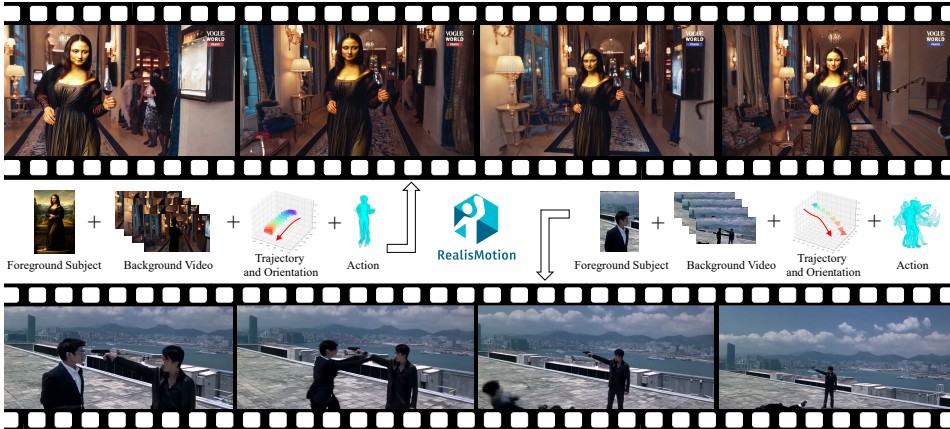

Figure 1: By decomposing the human motion into trajectory and action, and video appearance into foreground subject and background video, the proposed RealisMotion generates natural human motion videos by placing the foreground subject in the background video and having it perform the corresponding action along the specified trajectory. **We provide more than 100 video examples in the project homepage.**

In this paper, we introduce a decomposed human motion control and video generation framework that overcomes the limitations described above. Our key idea is to treat subject, background, trajectory, and action as independent, composable dimensions. This decomposition is realized in two stages. In the first stage, we represent human motion with the 3D parametric SMPL-X model Pavlakos et al. (2019) and build a 3D world coordinate system with physical ground awareness. After freely editing the 2D image-space trajectory, we unproject it into the 3D world space using depth estimation, focal-length calibration and coordinate transformation. The moving speed and human orientation are also aligned with the real motions. Then, the corresponding action sequence is retrieved from a motion bank or synthesized with text-to-motion methods. Finally, we render depth, normal, and color maps from the 3D scene to serve as conditioning guidance for subsequent video synthesis. In the second stage, we fuse these elements into coherent videos with a video generation model based on WAN-2.1 Wang et al. (2025). Starting from WAN-2.1-T2V, we fine-tune the model end-to-end with three key extensions: (1) subject injection via token concatenation along the sequence dimension, (2) background incorporation by channel-wise concatenation, and (3) motion (*i.e.*, trajectory + action) conditioning implemented with an additional ControlNet-style Zhang et al. (2023) module.

The contributions of this paper are summarized as follows.

1. We present a decomposed human motion-control and video-generation framework that models subject, background, trajectory, and action as independent, composable elements, enabling flexible mix-and-match editing. A detailed controllability comparison of related works is provided in Table 1.

2. We combine 3D physical priors with a learned video diffusion prior. The physical priors handle geometry-sensitive tasks (*e.g.*, 3D trajectory and action control, occlusion, and fore-shortening) in the 3D domain, while the video diffusion prior handles appearance and temporal aspects (*e.g.*, object/background control, frame consistency, and human–environment interaction) in the video domain.

3. We perform all trajectory and action edits in the 3D world space, preserving realistic speed, orientation, motion style and perspective effects.

4. We introduce a motion-conditioned video generation model built on the latest diffusion-transformer model Wan-2.1. Experiments on benchmark datasets and real-world cases show improved fidelity and controllability compared to prior motion-transfer methods.

## 2 METHOD

### 2.1 OVERALL PIPELINE

Given a reference human subject image $I$, a reference background video $V_{bgd}^{1:N}$, a sequence of target translation $T^{1:N}$ (also known as global trajectory), a sequence of target orientation $O^{1:N}$ and a

Table 1: Controllability comparison of related methods on four key video elements: trajectory (orientation reported separately for clarity), action, subject and background. ✓ denotes standalone and accurate control, while ✗ indicates limited, inaccurate, or joint control.

| Class | Example Methods | Trajectory | Orientation | Action | Subject | Background |
|---|---|---|---|---|---|---|
| T2V/I2V Base Models | Wan-2.1 Wang et al. (2025), *etc*. | ✗ | ✗ | ✗ | ✗ (joint) | ✗ (joint) |
| Image Animation | Animate Anyone Hu (2024), *etc*. | ✗ (2D) | ✗ (2D) | ✗ (2D) | ✗ (joint) | ✗ (joint) |
| | Tora Zhang et al. (2024c) | ✗ (2D) | ✗ | ✗ | ✗ (joint) | ✗ (joint) |
| Motion Control | MotionCtrl Wang et al. (2024d) | ✗ (2D) | ✗ | ✗ | ✗ (text) | ✗ (text) |
| | 3DTrajMaster Fu et al. (2024) | ✓ (3D) | ✓ (3D) | ✗ | ✗ (text) | ✗ (text) |
| **RealisMotion** (ours) | | ✓ (3D) | ✓ (3D) | ✓ (3D) | ✓ (image) | ✓ (image) |

sequence of target body pose $P^{1:N}$ (also referred to as human action), the goal in this paper is to generate a new video of the reference human moving in the background, following the defined motion (including $T^{1:N}$, $O^{1:N}$ and $P^{1:N}$). $N$ is the number of frames.

To achieve the goal, we first match the motion with the background in Sec. 2.2. Given the environment defined by the background video, the motion should follow the physical laws to ensure it appears reasonable and natural. Then, in Sec. 2.3, we propose a motion-guided video generation model that supports separate subject, background and motion control. By this two-stage design, we combine the 3D physical prior with the learned video diffusion prior for generating highly realistic human motion videos. We solve the 3D-related problems, such as 3D trajectory control, 3D global orientation control, 3D action control, occlusion and foreshortening, in the 3D domain; and we solve the rest problems, such as object control, background control, detail authenticity, frame consistency, human-environment interaction, motion error repairing, in the video domain.

## 2.2 DECOUPLED MOTION EDITING

### 2.2.1 MOTION REPRESENTATION

We use the SMPL-X Pavlakos et al. (2019) model for human body modelling in the low-level parametric space. It represents the human body as a function $\mathcal{M}(\gamma, \phi, \theta, \beta, \theta_h, \phi_f)$, which is parametrized by the global translation $\gamma \in \mathcal{R}^3$, global orientation $\phi \in \mathcal{R}^3$, body pose $\theta \in \mathcal{R}^{21 \times 3}$, body shape $\beta \in \mathcal{R}^{10}$, hand pose $\theta_h \in \mathcal{R}^{2 \times 15 \times 3}$ and facial expression $\phi_f$. After standard linear blend skinning and learned blend shape correction, the SMPL-X model outputs a 3D mesh representation with 10, 475 vertices. Hence, human motion could be well presented by a sequence of SMPL-X parameters.

To fit the motion into the background video, we need to make sure that both the motion and the environment in the background share the same 3D coordinate system: same coordinate origin, same axis direction and same coordinate scale. To avoid ambiguity, we build a world-grounded 3D coordinate system $(\overrightarrow{o}, \overrightarrow{x}, \overrightarrow{y}, \overrightarrow{z}, s)$ in the physical world without the impact of camera views in videos. More specifically, based on the human mesh recovery method GVHMR Shen et al. (2024), we define the coordinate system as follows: (a) the coordinate origin $\overrightarrow{o}$ is defined as the point where the human stands in the first frame of the video; (b) the $y$-axis $\overrightarrow{y}$ aligns with the gravity direction in the physical world; (c) we define the $x$-axis $\overrightarrow{x}$ and $z$-axis $\overrightarrow{z}$ as $\overrightarrow{x} = \overrightarrow{y} \times \overrightarrow{c}$ and $\overrightarrow{z} = \overrightarrow{x} \times \overrightarrow{y}$, respectively, where $\overrightarrow{c}$ is the camera view direction. In fact, it is difficult to align $\overrightarrow{x}$ and $\overrightarrow{z}$ for different $\overrightarrow{c}$, but we found that the x-z plane will always align with the ground plane given the definition of $\overrightarrow{o}$ and $\overrightarrow{y}$. Therefore, we can omit the mismatch of motion and environment in terms of $\overrightarrow{x}$ and $\overrightarrow{z}$, and rotate the 3D mesh with the rotation angle $\alpha$ between these two coordinate systems; (d) the coordinate scale $s$ is aligned with the physical distance, which means that a distance $d = 1$ in the coordinate system means 1 meter in the physical world.

### 2.2.2 TRAJECTORY AND GLOBAL ORIENTATION EDITING

With the SMPL-X model, we can directly change its parameters $\gamma$ and $\phi$ to control the trajectory $\Gamma^{1:N}$ and global orientations $\Phi^{1:N}$, where $N$ is the length of points in the given trajectory. Since editing these 3D parameters manually frame-by-frame is labor-intensive, we propose to first obtain the 2D trajectory, and then derive the 3D trajectory and the corresponding orientations based on two reasonable assumptions: (a) the human moves on the ground; (b) the human faces the direction of movement. The 2D points can be easily obtained by dragging the cursor or by selecting a few key points and applying linear interpolation.

Figure 2: The architecture of the proposed RealisMotion. It has two stages: 1) we first build a ground-aware 3D world coordinate system for the human motion, and conduct trajectory and action editing separately within the 3D space. 2) we then generate human videos conditional on the foreground subject image, background video and rendered motion guidance videos.

Formally, given a 2D point $\gamma_{2d}^n$ from the trajectory $\{\Gamma_{2d}^1, \Gamma_{2d}^2, ..., \Gamma_{2d}^N\}$ on the image, we represent it as $\Gamma_h^n$ in the homogeneous 2D image coordinates and unproject it to the 3D camera space as

$$\Gamma_c^n = K^{-1}\Gamma_h^n \cdot d * f_2/f_1 \tag{1}$$

where $K$ and $f_1$ are the camera intrinsic matrix and focal length predicted by GVHMR. $d$ and $f_2$ are the depth and focal length estimated by Depth Pro Bochkovskii et al. (2024). Here, we use $f_2/f_1$ for calibration as GVHMR only predicts a fake focal length according to the image size, which might lead to inaccurate transformations during motion editing.

Then, we further transform the 3D point $\Gamma_c^n$ from the camera space to the defined world space as

$$\Gamma_w^n = (\Gamma_c^n - T_{w2c})R_{w2c}^{-1} \tag{2}$$

where $R_{w2c}$ and $T_{w2c}$ are the rotation matrix and translation vector from the world space to the camera space. $R_{w2c}$ and $T_{w2c}$ are calculated based on the rigid point registration Umeyama (1991) of 3D human points between the world space and camera space in the background video.

Next, to make sure that the human moves with natural speed on the edited trajectory, we align the speed of the edited trajectory with the original speed. Otherwise, motion flaws such as feet sliding may occur when the feet move forward instead of maintaining static contact with the ground as would be expected in natural human motion. In detail, the alignment process starts with accumulating the total moving distance $\Delta^n$ from the first frame to the $n$-th frame as

$$\Delta^n = \sum_{i=2}^n \|\Gamma_w^i - \Gamma_w^{i-1}\|_1 \tag{3}$$

where $\|\cdot\|_1$ means the $\mathcal{L}_1$ norm. When we fit $\Delta^n$ and the edited translation $\Gamma^n$ as a function $\Gamma^n = \mathcal{F}(\Delta^n)$ for $n = 1, ..., N$, we can obtain the aligned translation $\bar{\Gamma}^n$ as $\bar{\Gamma}^n = \mathcal{F}(\Delta'^n)$, where the original total moving distance $\Delta'^n$ is defined similarly to $\Delta^n$ for the original trajectory.

After editing the trajectory, we edit the global orientation accordingly. For each frame $n$, we obtain the rotation angle $\Psi^n$ on the $x$-$z$ plane and derive the rotation matrix $R_n$ as

$$\Psi^n = atan(\frac{z^n - z^{n-1}}{x^n - x^{n-1}}), R^n = \begin{bmatrix} cos(\Psi^n) & 0 & -sin(\Psi^n) \\ 0 & 1 & 0 \\ sin(\Psi^n) & 0 & cos(\Psi^n) \end{bmatrix}. \tag{4}$$

To change the human orientation, we found that directly modifying $\Phi_n$ leads to unnatural swinging movements. Therefore, we apply the trajectory and orientation transformations together on 3D human vertices $\mathcal{V}^n$ as

$$\bar{\mathcal{V}}^n = (\Phi^n)^{-1}(\mathcal{V}^n - \Gamma^n)R^n + \bar{\Gamma}^n \tag{5}$$

Notably, due to the estimation errors, the edited human motion might suffer from feet floating or penetration to the ground. We shift vertices along the $y$-axis by subtracting the minimum $y$ value over a local temporal window to optimize foot contact. Besides, to improve motion consistency across frames, we also smooth the rotation angle in a sliding way during orientation editing.

### 2.2.3 BODY POSE AND HAND POSE EDITING

For body pose and hand pose, we can directly copy them from existing SMPL-X parameters. Consequently, we can easily collect a motion bank from existing videos with extracted SMPL-X parameters. When we use the motion to generate new videos, we just need to edit the trajectory and orientation according to the background, while the body pose and hand pose are kept unchanged. This allows us to retrieve different actions, such as walking, running and swimming, with their original action styles, from the motion bank. For repetitive motions, one can cut a clip of motion and repeat it as needed. As for the editing of body pose and hand pose, it is out of the scope of this paper and the readers can refer to related research such as Agrawal et al. (2023); Li et al. (2024).

In practice, the hand orientation $\Phi_h^n$ and hand pose $\Theta_h^n$ are estimated with an extra hand mesh recovery method HaMeR Pavlakos et al. (2024). It uses the parametric hand model MANO Romero et al. (2022) and estimates the hand parameters in the camera space. To match the hand with the human body in the world space, a quick solution is to match the HaMeR hand vertices with the SMPL-X hand vertices using rigid point registration, but it might result in incorrect waist rotations when the hand pose is significantly different from the standard hand pose of SMPL-X. Hence, we match the hand orientation parameters between MANO and SMPL-X by first reversing the original SMPL-X hand orientation and then apply the MANO orientation after camera-world space transformation. This is formulated as

$$\bar{\Phi}_h^n = (\Omega^n)^{-1}(\Phi_h^n R_{w2c}^{-1}) \tag{6}$$

where $\Omega^n$ is the hand orientation derived from the SMPL-X model using forward kinematics.

### 2.2.4 2D GUIDANCE RENDERING

Given the 3D human mesh representation, we render 2D depth maps, normal maps, and color maps to guide the video generation process. The same extrinsic and intrinsic camera parameters as the background video are used to ensure that the guidance maps and the target video are spatially aligned. In particular, the depth maps depict the distances from the camera to each pixel, while the normal maps contain the surface orientations of the meshes. Both of them provide critical geometric information for reconstructing the 3D structure of the human. Similar to RealisDance Zhou et al. (2024), we generate color maps by assigning different colors to different vertices, which can provide semantic information for different parts of the human, and improves human consistency across different frames. We also refer to RealisDance for rendering the hand maps. One thing to note is that we need to mask the occluded hand by comparing the depths of human body and hand. In addition, after we transferring motion from one human to the reference human subject, we use the body shape parameters $\beta$ of the reference subject, which allows us to keep the same body shape such as height and figure. When transferring motion from adults to children, we add an extra shape parameter to interpolate between SMPL-X and SMIL-X templates Patel et al. (2021); Hesse et al. (2018).

### 2.3 DECOMPOSED HUMAN VIDEO GENERATION

We build our human video generation model based on the text-to-video model Wan-2.1 Wang et al. (2025), which achieves state-of-the-art performance on video generation. It compresses the video into the latent space with a spatio-temporal causal variational autoencoder (VAE) Esser et al. (2021) and employs full attention Vaswani et al. (2017); Peebles & Xie (2023) for spatio-temporal contextual modeling of video tokens. As shown in Fig. 2, we decompose the video into several key elements for flexible and separate control, including foreground subject, background video, motion guidance and text.

**Subject Control** To control the subject, we first compress the subject image as image tokens using the Wan-2.1 VAE. Then, the image tokens are concatenated with the video tokens for full attention. To discriminate between reference image and target video tokens, we treat the reference image as a sufficiently distant video frame in the target video (for example, the 80-th frame) and

apply the corresponding rotary position embeddings (RoPE) Su et al. (2024) on it. This leads to a sufficiently large distance between image and video tokens during attention, while keeping the spatial composition of the reference image. In addition, we found the generated human face might be blurry possibly due to the fact that the face often occupies a relatively small area of the whole image. To improve the face performance, we detect the face in the reference image and upscale it as an extra reference image input. An ID embedding module similar to the time embedding module in Wan-2.1 is proposed for distinguishing the reference subject image and face image.

**Background Control**  To control the background of video, it is straightforward to compress the reference background as video tokens and then concatenate it with the target video tokens along the channel dimension, as the background video and the target video are supposed to be fully aligned. Typically, we obtain the background video with a human in it, especially in training. To avoid information leaking, we mask the foreground human in the background video with a mask. We also concatenate the mask with the video tokens along the channel dimension for helping the model identify the foreground area. In training, we additionally add random masks to background video to tackle with possible discrepancy between the target human area and masked foreground area during inference.

**Motion Control**  Given the rendered motion guidance videos, we encode them as visual tokens by VAE. Then, inspired by ControlNet Zhang et al. (2023), we copy the transformer blocks $\mathcal{T}$ of Wan-2.1 as $\mathcal{T}'$ and extract motion features $\mathbf{c}$ from different blocks. Next, we add the motion features to the video features $\mathbf{x}$ at corresponding positions for controlling the video motion. This is formulated as

$$\mathbf{c}^{b+1} = \mathcal{T}'^n(\mathbf{c}^b), \ \ for \ b = 1, ..., B \tag{7}$$

$$\mathbf{x}^{b+1} = \mathcal{T}^n(\mathbf{x}^b) + \mathcal{S}(\mathbf{c}^{b+1}), \ \ for \ b = 1, ..., B \tag{8}$$

where $b$ is the block index in $B$ blocks and $\mathcal{S}$ is a linear layer with zero initialization. To reduce model size and computation burden, we only use $B'$ blocks for motion feature extraction and add them to their neighboring blocks within a window size of $B/B'$. In other words, every $B/B'$ blocks share the same motion feature.

**Text Control**  It seems that a combination of the subject image, background video and driving motion can define a video well. However, we found that providing the text is still important for improving the model performance, possibly due to two reasons. First, the Wan-2.1 model was trained for the text-to-video task. Removing the text-related modules or providing empty text might lead to significant domain gaps. Second, there are still some undefined elements in the video, such as the other side of the reference human subject, or the interaction of human and environment. Therefore, we keep the text modules and annotate the video with corresponding text prompts. Particularly, we avoid the cross attention between the reference image tokens and text tokens in text modules, as we observe a performance drop of reference ID preservation ability.

**The Image-to-Video Variant**  We can seamlessly extend our model to the Wan-2.1 I2V (image-to-video) model, which additionally inputs the first frame of the video as a guidance. In this case, our model degenerates to be an image animation model when the reference subject and background are merged into a single image. It no longer supports separate subject-background customization, nor does it offer dynamic background control ability. We notice that there is a concurrent image animation work RealisDance-DiT Zhou et al. (2025), which could be adopted as our I2V variant to prevent duplicate efforts.

## 3 RELATED WORK

### 3.1 MOTION ACQUISITION

To generate human motion, one can directly estimate human motion by motion capture systems, which are often prohibitively expensive. With advancements in human motion recovery techniques, extracting human motion from images or videos has become significantly simpler and more accessible Kanazawa et al. (2018); Goel et al. (2023); Shin et al. (2024); Wang et al. (2024b); Shen et al. (2024); Zhang et al. (2024a); Yin et al. (2024). These methods predominantly use learnable neural networks to directly predict the parametric human model parameters in SMPL Bogo et al. (2016); Loper et al. (2023) or SMPL-X Pavlakos et al. (2019). Most of them follow a multi-stage pipeline

that consists of human bounding box tracking, 2D human keypoint detection, image feature extraction, camera relative rotation estimation and SMPL parameter regression. According to the difference of used coordinate systems, above methods can be roughly divided as camera-space Kanazawa et al. (2018); Goel et al. (2023); Zhang et al. (2024a) and world-space Shin et al. (2024); Shen et al. (2024); Wang et al. (2024b); Yin et al. (2024) methods. The former kind of method treats the camera as the origin and often fails to recover global motion due to accumulated translation and pose errors. In contrast, the latter kind of method defines a unified coordinate system without the impact of changing camera views, making it more suitable for subsequent motion editing.

Another way for motion generation is training generative models based on captured human motion datasets Punnakkal et al. (2021); Guo et al. (2022). Given different guidance, such as action label Cervantes et al. (2022), audio Aristidou et al. (2022) and natural language Ahuja & Morency (2019); Tevet et al. (2022a;b); Barquero et al. (2024), most methods choose conditional generative models to map from the conditioning domain to the motion domain. With significant advancements in diffusion models Sohl-Dickstein et al. (2015); Ho et al. (2020), many methods start to train diffusion models for human motions conditioning on texts Tevet et al. (2022b); Kim et al. (2023); Shafir et al. (2023); Barquero et al. (2024); Zou et al. (2024). For example, as one of the pioneering text-to-motion method, MDM Tevet et al. (2022b) adopts a transformer diffusion model for motion generation based on the CLIP text embedding.

## 3.2 Motion-Guided Video Generation

Similar to text-to-motion generation, diffusion-based models Blattmann et al. (2023); Zhu et al. (2023); Yang et al. (2024); Liang et al. (2024); Kong et al. (2024); Wang et al. (2025); Chen et al. (2025) have emerged as the current research mainstream for motion-guided video generation. As one of the pioneering methods, DisCo Wang et al. (2023) segments the foreground and background of the reference image, and then injects their VAE embeddings Esser et al. (2021) to the 2D UNet of Stable Diffusion Blattmann et al. (2023) by cross attention and ControlNet Zhang et al. (2023), respectively. The 2D pose sequence is encoded and injected into the UNet by ControlNet as well. As another representative method, Animate Anyone Hu (2024) upgrades the 2D UNet to a 3D UNet for better video quality. It also proposes a symmetric ReferenceNet to extract reference features, which are merged into the main network via spatial attention. The feature of 2D pose sequence is concatenated with the noise input for motion guidance. Subsequent methods basically follow the designs of DisCo and Animate Anyone, with improvements on base models Zhang et al. (2024b); Lin et al. (2025), reference injection Xu et al. (2024); Wang et al. (2024a); Zhou et al. (2025); Jiang et al. (2025), motion guidance Zhu et al. (2024); Tan et al. (2024); Men et al. (2024), hand fidelity Zhou et al. (2024), camera control Wang et al. (2024c); Shao et al. (2024), object interaction Hu et al. (2025), *etc.*. Some of them Zhu et al. (2024); Zhou et al. (2025) have used the SMPL models, but their exploration is limited to the camera space. It is worth pointing out that most above methods are essentially image animation methods, without any modification on extracted motions from existing videos. Artifacts might arise when the motion (generally represented in rendered 2D image space) mismatches with the reference image.

In particular, 3DTrajMaster Fu et al. (2024) attempts to control the object orientation and trajectory by representing them as the rotation-translation matrix, which is added with text embeddings to control video contents after cross attention. Since the non-rigid object motion is in fact defined by text prompts, it does not support complex and accurate motion control. Additionally, other techniques for modifying trajectories exist Yin et al. (2023); Wang et al. (2024d); Wu et al. (2024); however, the majority are limited to handling 2D rigid object movement and are not effective for intricate non-rigid human motion.

## 4 Experiments

### 4.1 Experimental Setup

Based on the Wan-2.1 14B model, we finetune our model on an internal dataset that comprises approximately 3,300 hours of multi-resolution human video content. The details are provided in the supplementary due to page limit. For evaluation, we compare our methods in several aspects. For trajectory and global orientation control, we compare the translation error and rotation error defined

Table 2: Comparison of trajectory and global orientation control with existing methods on the proposed Trajectory100 dataset.

| Method | Translation Error (m)↓ | Rotation Error (deg) ↓ | PSNR↑ | SSIM↑ | LPIPS↓ | FID↓ | FVD↓ |
|---|---|---|---|---|---|---|---|
| Wan-2.1-I2V Wang et al. (2025) | 10.349 | 0.418 | 14.96 | 0.4763 | 0.3260 | 33.06 | 1421.87 |
| Tora Zhang et al. (2024c) | 5.667 | 0.355 | 16.56 | 0.5195 | 0.2501 | 21.51 | 957.81 |
| RealisDance-DiT Zhou et al. (2025) | 1.706 | 0.167 | 16.17 | 0.4892 | 0.2481 | 23.02 | 758.08 |
| **RealisMotion** (ours) | **1.198** | **0.101** | **22.57** | **0.7664** | **0.0686** | **12.00** | **314.59** |

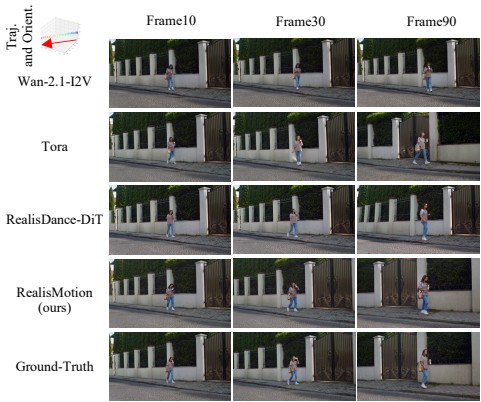

Figure 3: Visual comparison on trajectory and orientation control.

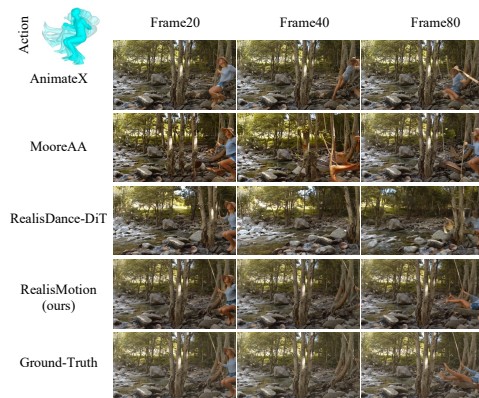

Figure 4: Visual comparison on action control.

by MotionCtrl Wang et al. (2024d), and also report video quality metrics including PSNR, SSIM, LPIPS Zhang et al. (2018), FID Heusel et al. (2017) and FVD Unterthiner et al. (2019). For action control, we mainly compare the video metrics with existing image animation methods.

## 4.2 COMPARISON WITH EXISTING METHODS

### 4.2.1 TRAJECTORY AND GLOBAL ORIENTATION CONTROL

To assess trajectory and global orientation control capabilities, we created a 100-video evaluation dataset with distinct movement paths, named Trajectory100. We compare our approach against the Wan-2.1 base model Wang et al. (2025), the trajectory-focused method Tora Zhang et al. (2024c), and the image animation method RealisDance-DiT Zhou et al. (2025). As illustrated in Table 2, our proposed RealisMotion outperforms all models in each metric. The lowest translation and rotation errors demonstrate superior trajectory and global orientation control, while additional metrics confirm that our generated videos also offer the highest visual quality. Fig. 3 shows that although Tora and RealisDance-DiT can control human trajectories in the 2D camera space to some extent, their outputs do not accurately represent physical positions within the environment. Furthermore, related methods like MotionCtrl Wang et al. (2024d) and 3DTrajMaster Fu et al. (2024) are excluded since their video backgrounds and objects are specified by text prompts, making quantitative evaluation on Trajectory100 difficult. A detailed comparison of controllability is available in Table 1.

### 4.2.2 ACTION CONTROL

We evaluate the action control performance of various methods using the image animation benchmark dataset RealisDance-Val Zhou et al. (2025). As presented in Table 3, RealisMotion significantly surpasses existing methods across all five metrics, demonstrating its robust action control capabilities. The qualitative results, depicted in Fig. 4, reveal that our approach produces clear, visually appealing videos with accurate actions, whereas the comparative methods often result in unnatural, distorted human figures.

### 4.2.3 SUBJECT AND BACKGROUND CONTROL

As depicted in Fig. 1 and Fig. 5, our approach allows for arbitrary subject customization and movement within existing background videos by referring to a reference image. Although our model has been mainly trained on adult human videos, it demonstrates strong generalization capabilities to previously unseen animation characters and children. In terms of background control, the effectiveness of our approach is illustrated in the last two rows of Fig. 3 and Fig. 4, wherein it consistently

| Method | PSNR↑ | SSIM↑ | LPIPS↓ | FID↓ | FVD↓ |
|---|---|---|---|---|---|
| Animate-X Tan et al. (2024) | 16.29 | 0.5893 | 0.2664 | 36.50 | 2376.66 |
| ControlNeXt Peng et al. (2024) | 15.66 | 0.5762 | 0.2776 | 40.38 | 2412.52 |
| MimicMotion Zhang et al. (2024b) | 17.20 | 0.6029 | 0.2457 | 43.51 | 2283.93 |
| MooreAA Hu (2024) | 16.08 | 0.5546 | 0.2488 | 37.92 | 2446.50 |
| MusePose Tong et al. (2024) | 17.29 | 0.6080 | 0.2276 | 44.66 | 2809.02 |
| RealisDance-DiT Zhou et al. (2025) | 17.22 | 0.5919 | 0.2050 | 26.18 | 1576.66 |
| **RealisMotion** (ours) | **20.34** | **0.7224** | **0.0998** | **20.67** | **1000.98** |

Table 3: Comparison of action control on RealisDance-Val Zhou et al. (2025).

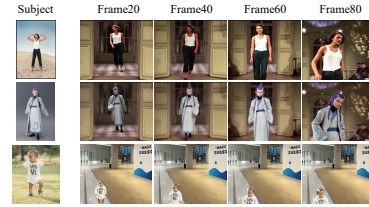

Figure 5: Visual results of subject control.

preserves background continuity, a feature not observed in the comparative methods. Note that the recent Animate Anyone 2 Hu et al. (2025) is not compared here as it is not open-sourced.

## 4.3 ABLATION STUDY

We conduct ablation study on Trajectory100. The accompanying visual comparison and additional ablation studies are provided in the supplementary.

**Focal Length Calibration**  To mitigate the adverse effects of inaccurate focal length, we calibrate the focal length. As demonstrated in Table 4, the PSNR decreases from 22.57dB to 21.52dB when calibration is absent. Visual examples in the supplementary material reveal that, without calibration, the human size may appear inconsistent with the surrounding environment, thereby contravening physical commonsense.

Table 4: Ablation Study on different designs. The accompanying visual results are provided in the supplementary.

| Ablation Study (w/o) | PSNR↑ | LPIPS↓ |
|---|---|---|
| Focal Length Calibration | 21.52 | 0.1043 |
| Body Hand Matching | 22.34 | 0.0694 |
| Text Prompt | 22.12 | 0.0793 |
| Extra Face Input | 22.36 | 0.0701 |
| Shifted RoPE | 22.13 | 0.0752 |
| Random Masking | 21.88 | 0.0951 |
| **RealisMotion** (ours) | **22.57** | **0.0686** |

**Body-Hand Matching**  Given that the human body and hands are predicted using different methods and within different spaces, we align the hands with the body to achieve more precise hand pose control. In the absence of this alignment, the default hand pose is used, resulting in a decrease in PSNR to 22.34dB.

**Text Prompt**  Since the foreground, background, and motion effectively define a video, we attempt to remove the text module to reduce computational demands and simplify the inference process. However, as indicated in Table 4, this leads to a performance drop in video quality. The visual results provided in the supplementary reveal that the resulting videos tend to generate incorrect details.

**Shifted RoPE for Reference Subject Image**  We propose to shift the RoPE to differentiate between the reference image and the target video. Without this design, the PSNR decreases to 22.13dB. The visual results in the supplementary material show that the first frame deteriorates significantly, likely because the absence of RoPE on the reference frames actually causes the reference frame to be treated as the first frame.

**Extra Face Image for Reference**  With an additional face image input, the PSNR improves from 22.36dB to 22.57dB. This enhancement is further corroborated by the visual comparisons provided in the supplementary.

**Random Masking On Background**  We randomly apply masking to the background to address the mismatch between background and motion during inference. As illustrated in the supplementary, the absence of random masking can lead to the generation of two human figures: one in the original human region and another in the new motion region, resulting in significant performance drops, as indicated in Table 4.

## 5 CONCLUSIONS

In this paper, we present RealisMotion, a decomposed human motion control and video generation framework. It constructs a ground-aware 3D world coordinate system that enables straightforward, realistic trajectory and action editing in the 3D space. Using the rendered motion guidance, RealisMotion synthesizes videos with independent control over foreground subject, background, trajectory, and action. Extensive experiments demonstrate state-of-the-art video quality and superior controllability across these elements.

**Limitation and Future Work**  Currently, our method has limited sensitivity to the environment's 3D structure and can sometimes produce foreground–background lighting inconsistencies. We leave these challenges for our future work.

## 6 ETHICS STATEMENT

This work focuses on generating human videos with realistic and controllable motions, enabling flexible composition of subjects, actions, trajectories, and backgrounds. While our method opens up promising applications in virtual content creation, animation, and interactive systems, we recognize that video generation technologies can also pose ethical risks if misused. In particular, highly realistic synthetic videos may be used to create misleading or harmful content, such as deepfakes, without consent. To mitigate such risks, we emphasize that our framework is designed for controllable and transparent synthesis, where each component is explicitly specified by the user. We do not support or encourage the generation of content involving real individuals without their permission, nor do we intend for our method to be used in deceptive, discriminatory, or privacy-violating ways.

We also acknowledge concerns regarding bias and fairness in training data: motion patterns and appearances may reflect societal biases present in datasets. We encourage future work to incorporate fairness-aware data curation and evaluation practices. Finally, we affirm that this research was conducted in accordance with principles of research integrity, and no human subjects were involved without appropriate oversight or informed consent.

## 7 REPRODUCIBILITY STATEMENT

We are committed to ensuring the reproducibility of our results. To this end, we provide comprehensive details of our method in the main paper and appendix, including model architecture, training procedures, hyper-parameters, and evaluation protocols. For full transparency, we include an anonymized version of our source code as supplementary materials.

## 8 DISCLOSURE OF LARGE LANGUAGE MODELS USAGE

We used GPT-4o to assist with language editing and proofreading, including improving the clarity, grammar, and style of the manuscript. The model was not involved in generating technical content, designing experiments, or producing results. All ideas, analyses, and conclusions are the sole responsibility of the authors.

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

# A   APPENDIX

In this supplementary material, we begin by detailing the data processing and training procedures in Sec. B. We then present additional visual results in Sec. C, along with visual comparisons from the ablation studies in Sec. D. Subsequently, we include further ablation studies concerning the proposed model in Sec. E. Finally, we integrate our method with text-to-motion techniques and the I2V variant Realisdance-DiT, as discussed in Sec. F and Sec. G. We highly recommend readers visit our project homepage to view the video results.

# B   EXPERIMENTAL SETUP

In data preprocessing, we first estimate the human motion with hand pose by GVHMR Shen et al. (2024) and HaMeR Pavlakos et al. (2024), and generate captions for videos by LLaVA Liu et al. (2024). Then, based on the rendered video depths, we derive the human mask and use image erosion for expanding the mask region, due to that fact that the rendered depth region is often smaller than the real human region. The obtained mask is used for segmenting the reference human subject and background. During training, we randomly sample video clips and randomly select one frame as the reference human subject image. The driving motion videos are randomly masked to improve its robustness to unnatural motion. During inference, we generate the corresponding caption by requesting GPT-4o Hurst et al. (2024) to describe the human subject image and the first frame of the background video. Besides, MatAnyone Yang et al. (2025) is employed for better foreground background segmentation during inference.

During training, we use the AdamW optimizer Loshchilov & Hutter (2017) to train the model with the flow matching loss Lipman et al. (2022). Initially, we train the model for 20,000 iterations using 49-frame video clips at 8fps and 480p resolution, with a batch size set to 128. Then, we finetune it using 97-frame video clips at 16fps and 720p resolution, for another 20,000 iterations. The learning rate and weight decay are set as 1e-5 and 1e-4, respectively. Other noise scheduler settings are kept the same as Wan-2.1. In particular, we randomly set the text and foreground subject to null or zeros for the purpose of classifier-free guidance (CFG) in inference. On a server with 8 H20 GPUs, it takes about 20 minutes to generate a $97 \times 1280 \times 720$ video (about 6 seconds).

# C   MORE VISUAL COMPARISON

We offer additional visual comparisons of human video generation with existing methods in Fig. 6, Fig. 7, and Fig. 9. These figures clearly demonstrate that our method outperforms all competing approaches. We also show more multi-resolution examples on trajectory, action, subject or background editing of our method in Fig. 8.

# D   VISUAL COMPARISON FOR ABLATION STUDY IN THE MAIN PAPER

## D.1   FOCAL LENGTH CALIBRATION

As illustrated in Fig. 10, the human size may not align correctly with the surrounding environment, appearing too large when perceived as too close or too small when viewed as distant. This discrepancy contravenes the principles of physical commonsense.

## D.2   BODY-HAND MATCHING

As depicted in Fig. 11, we compare the results of no matching, rigid registration-based matching, and our proposed method. The rendered hand guidances are presented in the top section, while the resulting video is displayed in the bottom section. It is clear that our approach effectively aligns the body and hand with precision, whereas the compared methods either revert to a default hand pose or encounter inaccurate wrist angles. These issues result in incorrect hand poses and flawed hand-object interactions in the videos.

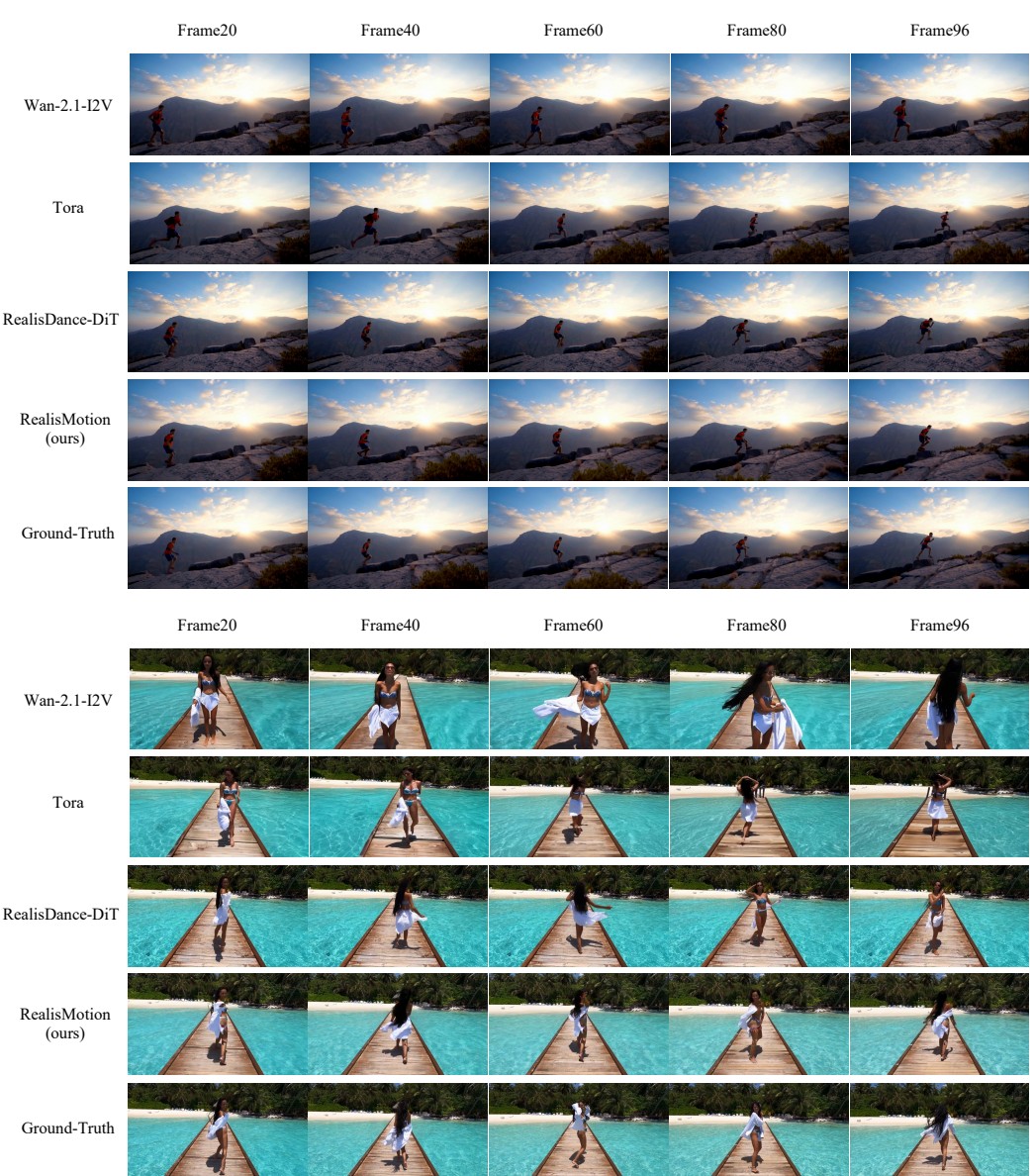

Figure 6: More visual comparison of different methods on trajectory and global orientation control. The video results are provided in the project homepage.

### D.3 TEXT PROMPT

As illustrated in Fig. 10, removing text-related modules results in diminished video quality and inaccuracies in details described by the text. For example, in the sample video, the woman in the ground-truth is wearing a jumpsuit, whereas the generated video incorrectly depicts her wearing long socks.

### D.4 SHIFTED ROPE FOR REFERENCE SUBJECT IMAGE

From Fig. 10, it is evident that without shifted RoPE, the video suffers from severe artifacts, particularly in the first and last frames. The first frames exhibit strips around the human regions, while the last frames show frog-like artifacts surrounding the human areas in the generated video.

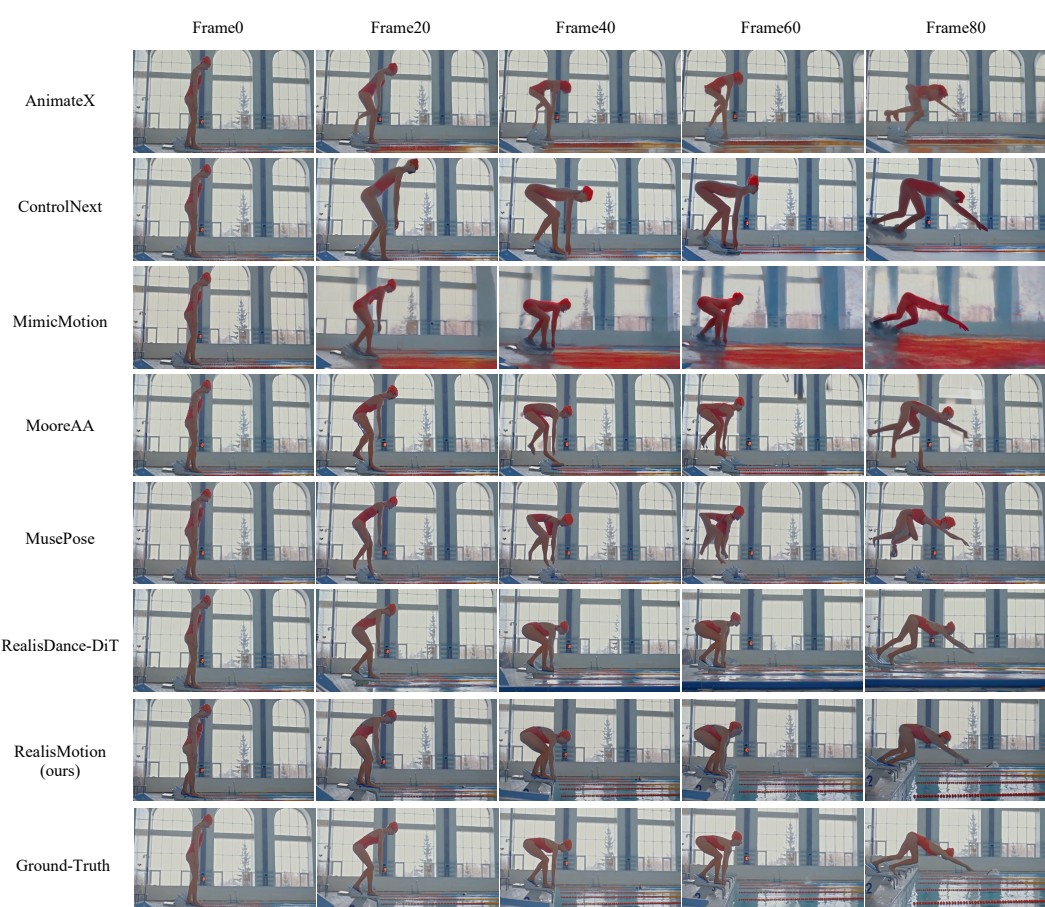

Figure 7: More visual comparison of different methods on action control. The video results are provided in the project homepage.

## D.5 EXTRA FACE IMAGE FOR REFERENCE

Incorporating an additional face image as model input enhances the face quality in the generated video, as demonstrated by the comparisons in Fig. 10. The faces generated by competing methods without extra face guidance tend to be more blurry.

## D.6 RANDOM MASKING ON BACKGROUND

As depicted in Fig. 12, no random masking during training leads to difficulties for the proposed method in accurately identifying the correct human position, resulting in the presence of two figures in the produced video (one at the original human region). This issue likely arises because the background and motion are consistently aligned during training, a condition that frequently does not hold during inference.

## E MORE ABLATION STUDY

### E.0.1 TRANSFORMATION ON 3D HUMAN VERTICES

When altering the trajectory or global orientation, transformations can be directly applied to the SMPL-X parameters or alternatively to the rendered human vertices. As demonstrated in Fig. 13, from the hand and shoulder positions, we can conclude that applying transformations to the vertices

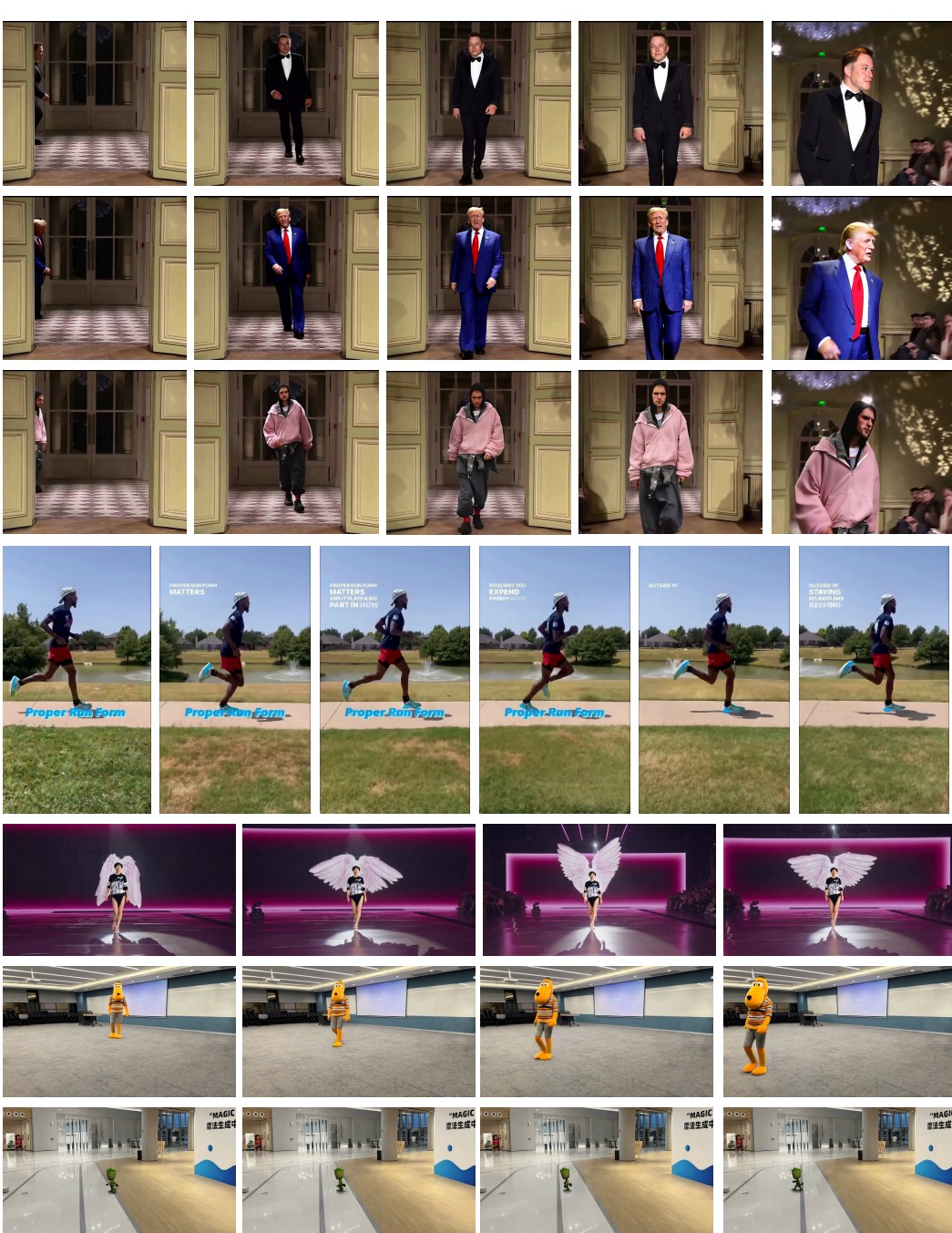

Figure 8: More visual results on trajectory, action, subject or background editing. The video results are provided in the project homepage.

results in more stable human motion and reduces shaking than on SMPL-X parameters. The shaking becomes more apparent when the sequence is viewed as a video.

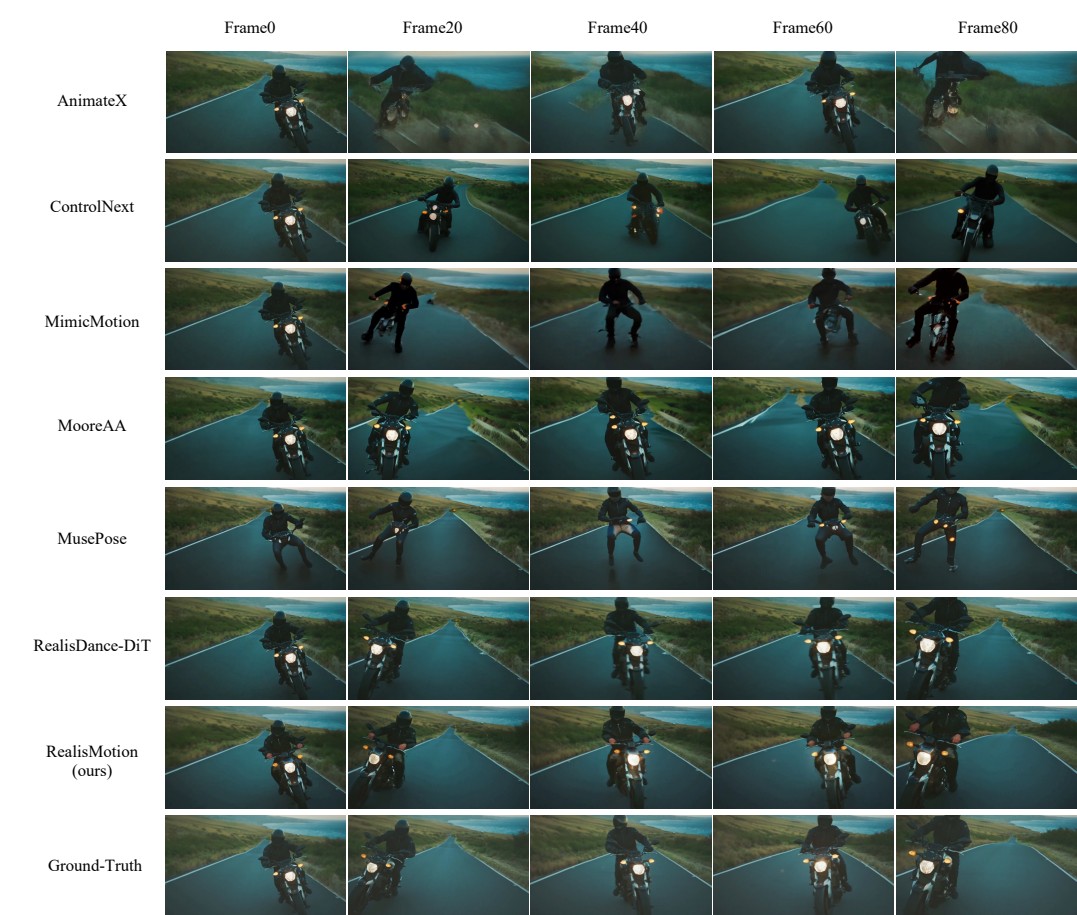

|  | Frame0 | Frame20 | Frame40 | Frame60 | Frame80 |

AnimateX

ControlNext

MimicMotion

MooreAA

MusePose

RealisDance-DiT

RealisMotion
(ours)

Ground-Truth

Figure 9: More visual comparison of different methods on action control. The video results are provided in the project homepage.

### E.0.2 THE IMPACT OF CAMERA-SUBJECT DISTANCE

As illustrated in Fig. 14, increased distances result in poorer visual details of the human subject. This is a common challenge for trajectory editing. This is because that the foreground human occupies a smaller area (only a few tokens) in the video, when it is positioned far from the camera.

### E.0.3 CLASSIFIER-FREE GUIDANCE

We utilize classifier-free guidance (CFG) for model inputs such as text and the foreground subject. As illustrated in Fig. 15, CFG applied to text enhances video details and reduces artifacts, while CFG on the foreground subject improves the ability to preserve human identity (such as face details and dresses). Furthermore, we observe that even without a foreground subject, the proposed method can still generate a coherent video from the text prompt. Essentially, our approach functions as a specialized video inpainting model guided by both the foreground subject and text. In scenarios where the background is absent, it becomes a specialized video outpainting model. When neither foreground subject nor background is provided, our method degrades to a human video generation model driven by text input and motion guidance.

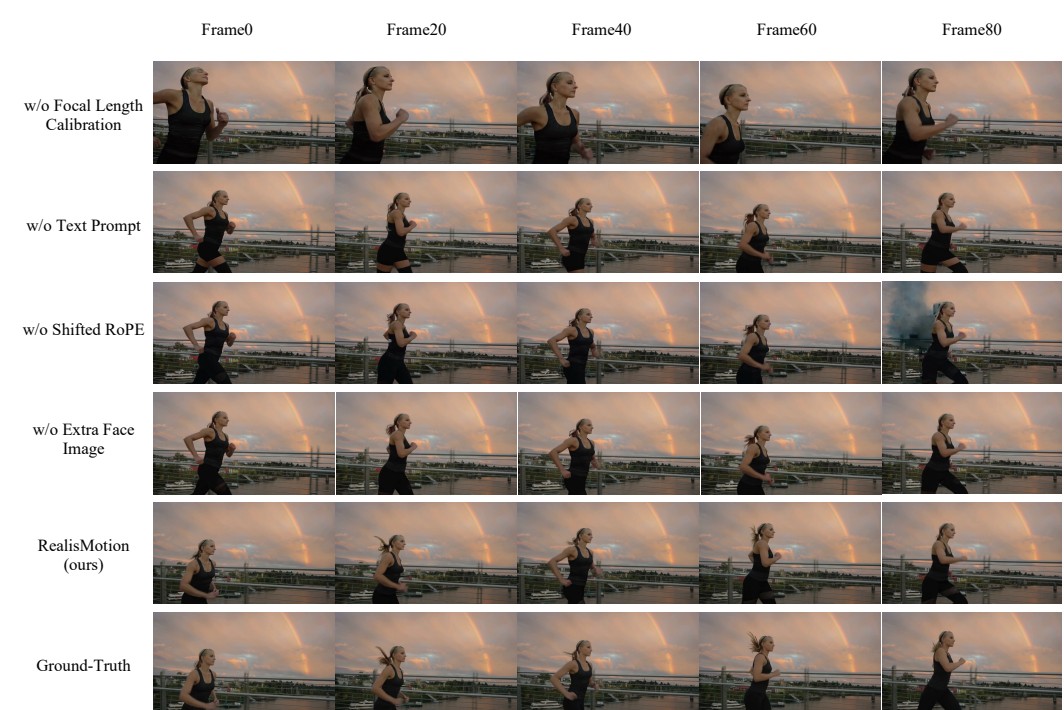

Figure 10: Visual results of several ablation studies. Please zoom in for better comparison.

## F    COMBINATION WITH TEXT-TO-MOTION METHOD FLOWMDM

Our proposed method could be seamlessly integrated with text-to-motion techniques. For instance, using the motion generated by FlowMDM Barquero et al. (2024), we can render the 2D guidance and produce the corresponding video, as illustrated in Fig. 16. This integration enables our method to leverage the motion editing capabilities of existing text-to-motion approaches as well.

## G    COMBINATION WITH THE I2V VARIANT REALISDANCE-DIT

RealisDance-DiT Zhou et al. (2025) can be considered an image-to-video variant of our method, defining the subject and background within a single image that serves as the first frame of the video. With edited motion, RealisDance-DiT can also be employed to generate the corresponding video. Some example results are presented in Fig. 17.

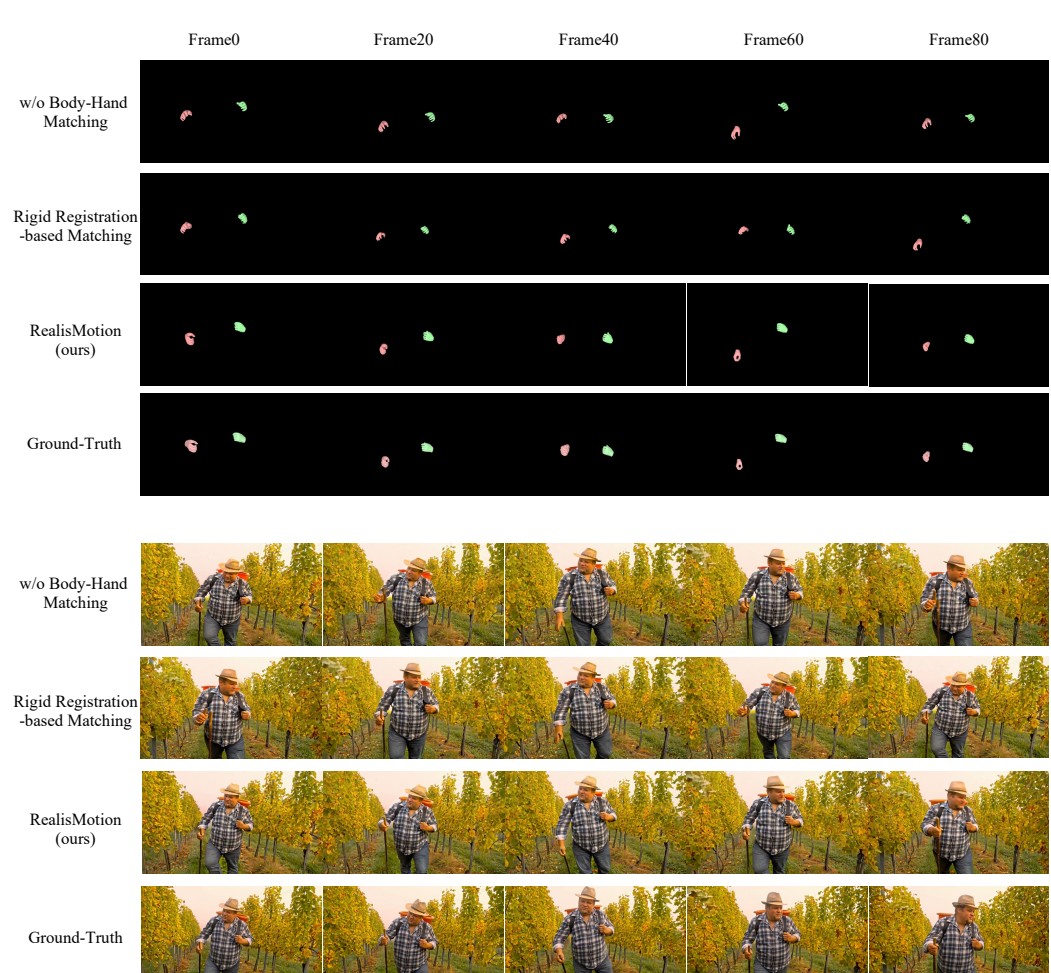

Figure 11: Visual results of ablation study on body-hand matching. Please zoom in for better comparison. The readers are suggested to focus on the human hands in this example.

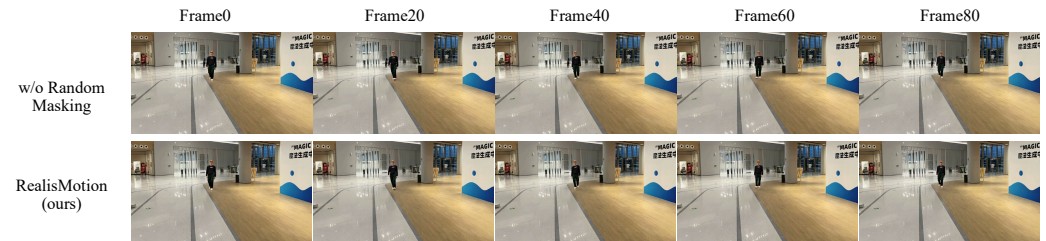

Figure 12: Visual results of ablation study on random masking. The artifacts appear to the left of the human figures in this case. Please zoom in for better comparison.

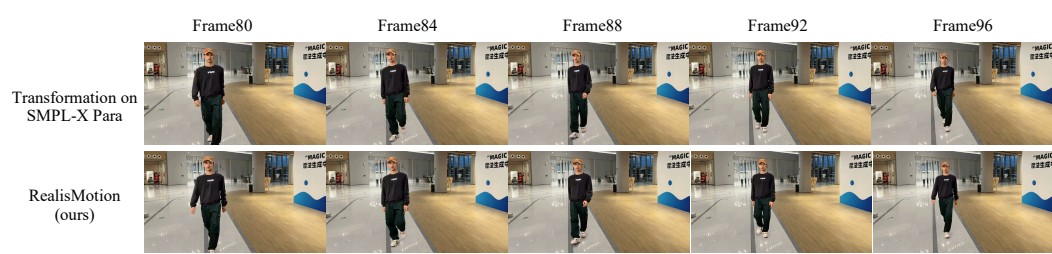

Figure 13: Visual results of ablation study on vertice transformation. The shaking of human motion is more visible when the sequence is viewed as a video. Please zoom in for better comparison.

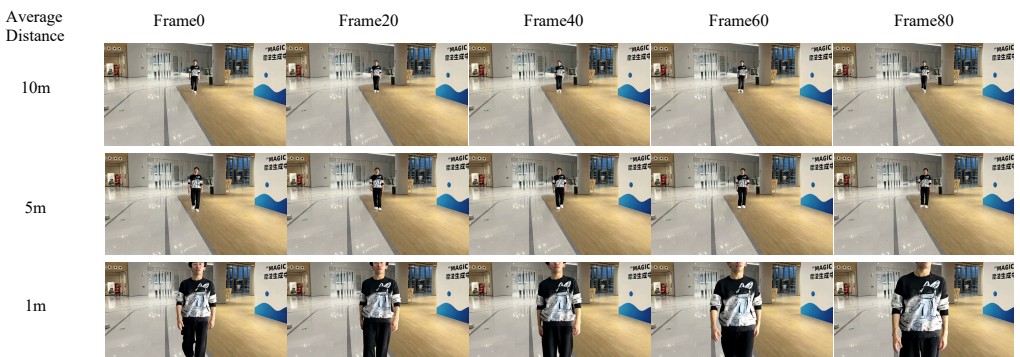

Figure 14: Visual results of ablation study on the vertice transformation. Please zoom in for better comparison.

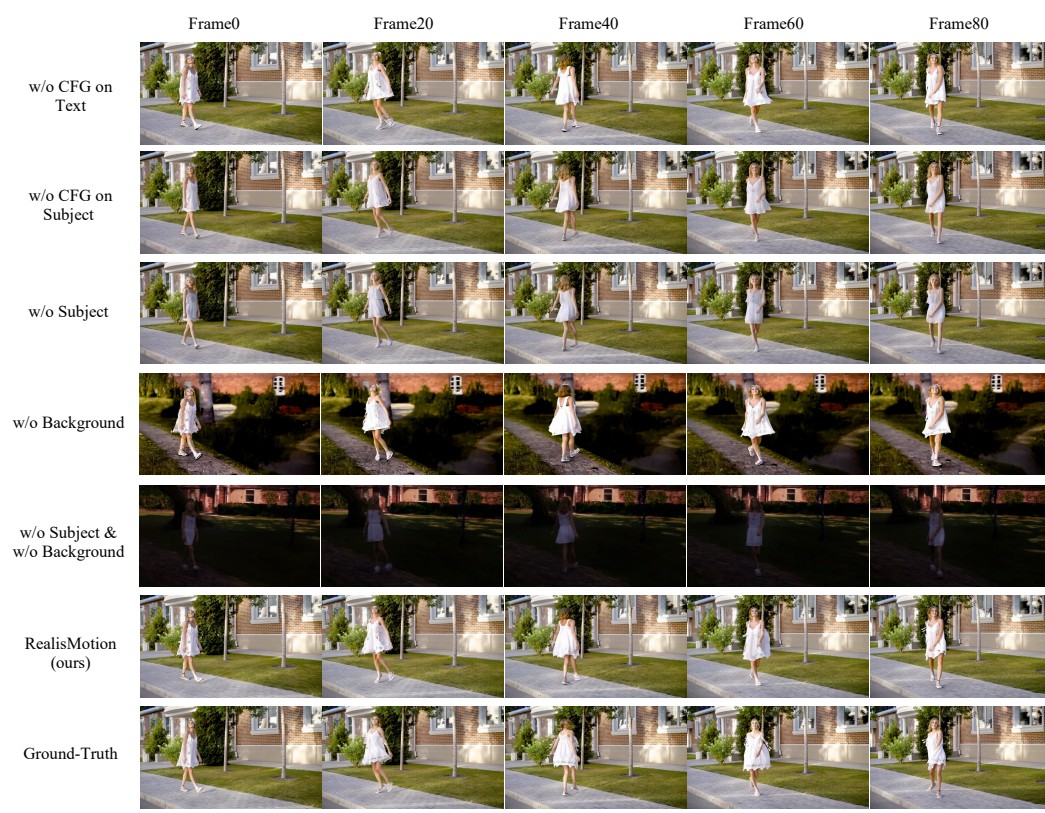

Figure 15: Visual results of ablation study on classifier-free guidance (CFG) for different inputs. Please zoom in for better comparison.

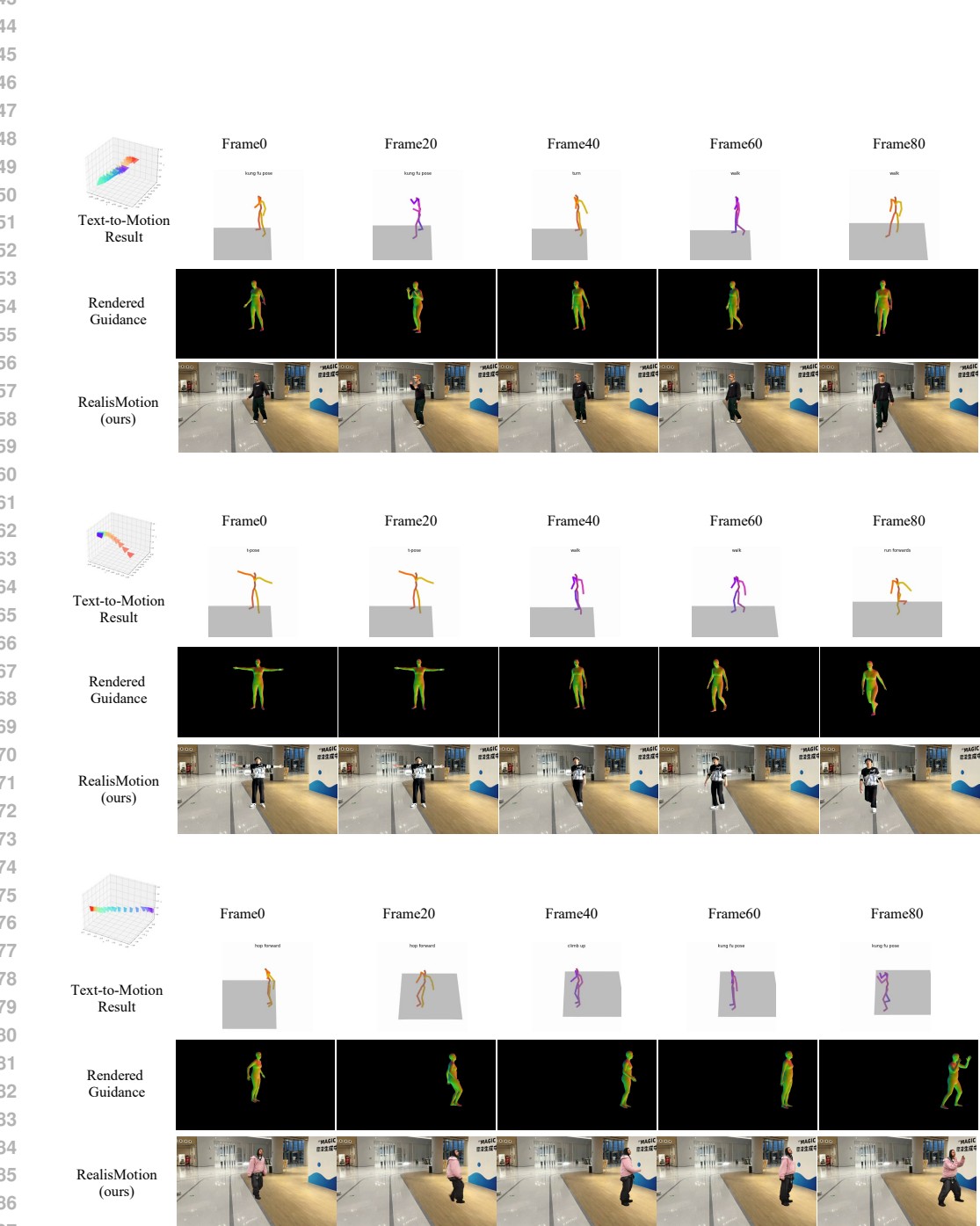

Figure 16: Visual results of combination with the text-to-motion method FlowMDM. Please zoom in for better comparison.

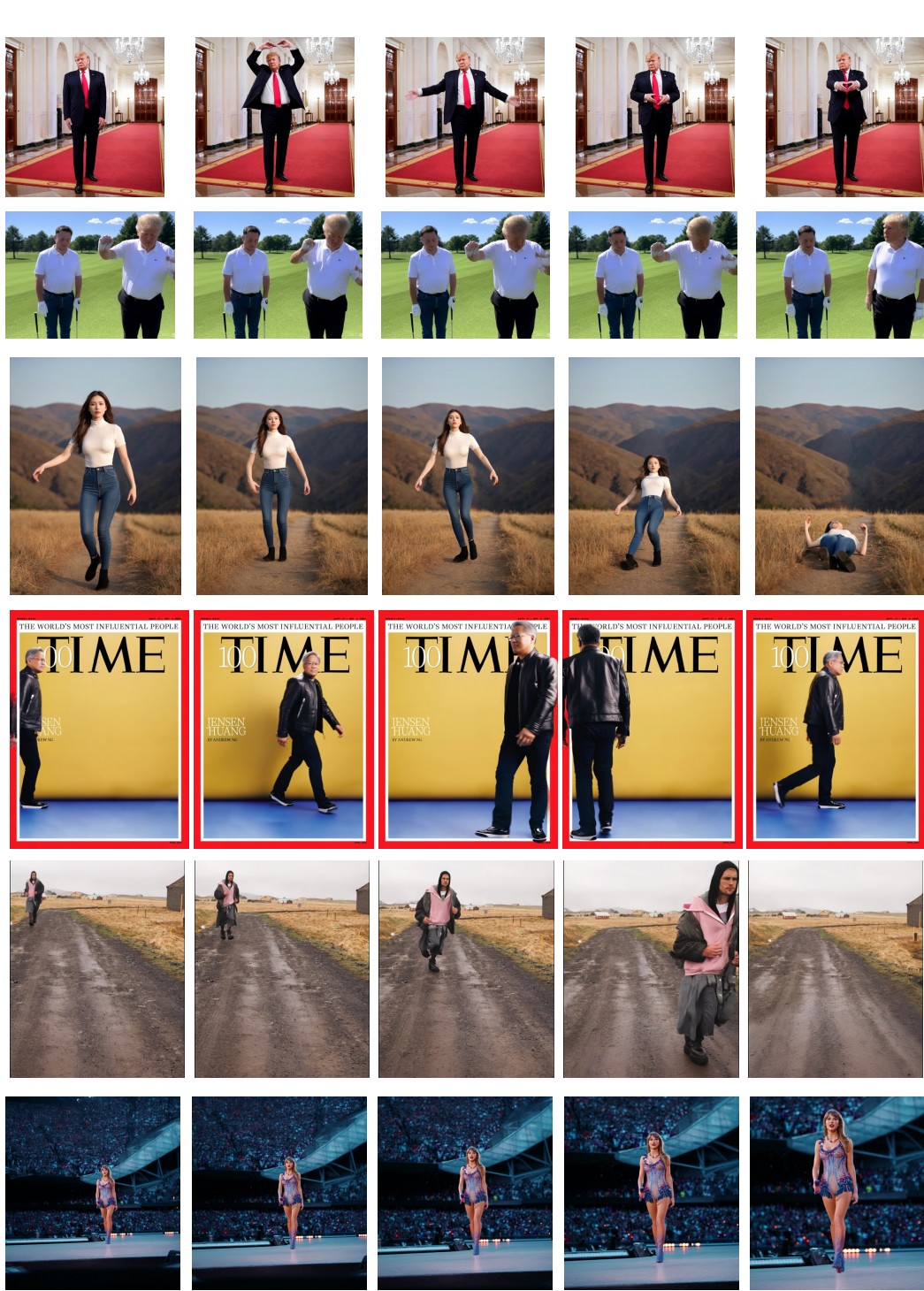

Figure 17: Visual results of combination with the I2V variant RealisDance-DiT. Please zoom in for better comparison.

