# OpenReview forum: "RealisMotion: Decomposed Human Motion Control and Video Generation in the World Space"
_ICLR.cc/2026/Conference — ICLR 2026 Conference Withdrawn Submission_

### Official Review · Reviewer_rBST · 2025-10-25

**Soundness:** 2
**Presentation:** 2
**Contribution:** 2
**Rating:** 4
**Confidence:** 3

**Summary:**

This paper introduces RealisMotion, a framework designed for realistic human video generation with decoupled control over four key elements: the foreground subject, the background video, the human trajectory, and the action pattern. The method operates in two main stages: decoupled Motion Editing, decomposed Video Generation. The method is evaluated on a new Trajectory 100 dataset and the RealisDance-Val benchmark, showing state-of-the-art performance in both controllability and video quality.

**Strengths:**

1. The paper tackles a central challenge in video generation: the full, independent control of subject, background, and motion.
2. The method successfully combines 3D physical priors with a learned video diffusion prior.
3. The method demonstrates clear state-of-the-art performance, achieving the lowest translation and rotation errors on its trajectory benchmark (Table 2) and the highest quality scores on the action control benchmark (Table 3).

**Weaknesses:**

1. The model is fine-tuned on an internal dataset of 3,300 hours of video. This is a major reproducibility flaw. It makes it impossible to distinguish whether the model's SOTA performance comes from the novel architecture or from this massive, proprietary dataset, which the baseline models did not use.
2. The primary benchmark for trajectory control, Trajectory 100, was created by the authors and is not public. This, combined with the internal training data, makes the SOTA claims difficult to verify.
3. The paper's claim of 3D trajectory control is limited. The editing process is 2D-first, and the 3D unprojection relies on two strong assumptions: the human moves on the ground and the human faces the direction of movement. This framework would fail for any actions involving jumping, swimming, or simply walking backward, which is a significant limitation on the doing anything claim.
4. The action component is primarily driven by a motion bank of pre-existing SMPL-X sequences. This limits actions to what has already been captured and processed.
5. The full pipeline is exceptionally complex, requiring numerous pre-processing models (GVHMR, Depth Pro, HaMeR, MatAnyone, LLaVA) on top of a 14B parameter generative model. The reported inference time of 20 minutes for a 6-second video is prohibitive.
6. The paper itself notes (in the appendix) that performance degrades significantly as the subject moves farther from the camera. This is a critical failure case for a method that explicitly allows free trajectory editing.
7. The authors state as a limitation that the method has limited sensitivity to the environment's 3D structure and can produce lighting inconsistencies. This breaks the realistic claim, as the subject often appears pasted onto the background without shadows, reflections, or other physical interactions.
8. The ablation study shows that Body-Hand Matching is critical. However, the paper's solution is complex, involving reversing SMPL-X orientation and applying MANO orientation. The appendix (Fig. 11) shows this is an improvement, but it's another complex, multi-model dependency (requiring HaMeR) to fix a problem in the base SMPL-X model.

**Questions:**

There are some aspects of this paper that need improvement, which are provided in the Weaknesses section. These primarily concern the fairness of the experiments and the limitations of the application. I am open to increasing my score if the authors can effectively address these concerns.

---

### Official Review · Reviewer_WgY7 · 2025-10-30

**Soundness:** 3
**Presentation:** 3
**Contribution:** 2
**Rating:** 4
**Confidence:** 4

**Summary:**

This paper proposes RealisMotion, a novel framework for human video generation that achieves fine-grained control by explicitly decomposing the video content into four independent, composable elements: foreground subject, background video, human trajectory, and action patterns. The key idea is to perform motion editing, including trajectory and action, in a ground-aware 3D world coordinate system before fusing the elements via a specialized Video Diffusion Transformer, which is built upon the WAN-2.1 architecture. This approach successfully decouples geometry-sensitive control from appearance and temporal consistency. Experimental results indicate the effectiveness of the proposed method.

**Strengths:**

1.	RealisMotion introduces an unparalleled level of explicit, independent control over four fundamental video elements, including subject, background, trajectory, and action.
2.	The framework successfully integrates 3D motion priors with modern video diffusion priors, i.e., WAN-2.1-T2V. Experimental results verify the effectiveness of the proposed method.

**Weaknesses:**

1.	This reliance on large-scale internal data creates a major hurdle for reproducibility and makes it impossible for the academic community to verify the results or build upon the fine-tuned model. The performance gains might stem more from the scale and quality of this undisclosed training data than from the architectural novelty of RealisMotion itself.
2.	The comparison against existing state-of-the-art methods (e.g., Animate Anyone, MotionCtrl, 3DTrajMaster) is inherently unfair if these methods are trained only on public datasets (e.g., public TikTok, standard motion datasets) while RealisMotion benefits from large-scale, potentially proprietary or industrial-scale internal video data.
3.	The entire 3D motion pipeline is predicated on the accuracy of the GVHMR human mesh recovery method for initial pose, camera parameters, and the estimation of 3D points. It also relies on Depth Pro for depth and focal length estimation. Failures or noise in these heavy upstream models will inevitably compromise the quality and consistency of the 3D control signals, limiting the robustness of the system in real-world, highly challenging scenarios.
4.	The inference process and the inference speed should also be included.

**Questions:**

How can the users obtain the control signals and use them to generate plausible results?

---

### Official Review · Reviewer_WxXm · 2025-11-01

**Soundness:** 3
**Presentation:** 3
**Contribution:** 3
**Rating:** 4
**Confidence:** 3

**Summary:**

This paper introduces RealisMotion, a 3D controllable video generation system comprising two main stages.
In the first stage, human motion, trajectory, and orientation are generated within a unified 3D world coordinate system.
The second stage uses the motion sequence rendered in Stage 1 as guidance, along with a reference image and a background video with a masked foreground subject, to produce the final human video. This framework treats character, background, motion, and trajectory as independently editable dimensions, enabling more precise and controllable video generation. Both quantitative and qualitative experiments demonstrate the method’s effectiveness compared to baseline approaches.

**Strengths:**

- The authors propose and implement a sophisticated 3D-conditioned video generation system. The paper is well-structured, mainly organized into two parts: 3D motion generation and editing, and 2D human video generation, making it easy to follow.
- The system incorporates four key inputs: (1) a reference image, (2) a background video, (3) target translation and orientation, and (4) a motion sequence, allowing independent control over these four dimensions.
- The paper systematically discusses how trajectory, orientation, and motion are generated, with careful consideration of coordinate system construction and alignment. It also addresses issues such as aligning the speed of the edited trajectory with the original motion speed.
- For the conditioned video generation, the authors adopt existing techniques, including reference injection (temporal concatenation), background injection (channel concatenation), and motion video injection (ControlNet).
- Quantitative and qualitative experiments demonstrate the superiority of the proposed method, and ablation studies validate the effectiveness of its individual components.

**Weaknesses:**

**Discussion on System Limitations and Trade-offs**

The system relies on 3D guidance for control and a video generation model to produce the final video. This introduces a natural trade-off between the richness of the 3D guidance and the generalization capacity of the video model, i.e., balancing 3D physical priors with video diffusion priors. For instance:

- Using detailed 3D meshes as guidance enhances body shape consistency but limits the diversity of expressions (e.g., it might struggle to depict a woman with long flowing hair in a dress).
- Using a less constrained guidance, such as skeletons, allows for more diverse clothing or body movements but may fail to represent entities with significant shape variations, such as monsters.

**Motion-Related Concerns**

1. Separation of Trajectory, Orientation, and Motion: Treating trajectory/orientation separately from motion could lead to inconsistencies. For example, a trajectory indicating forward movement paired with a stationary jumping-jack motion could result in unrealistic outcomes.
2. Handling Multi-person Motion: It is unclear how well the system can handle complex multi-person interactions, such as hugging or fighting motions.

**Background Control**

1. **Random Masking**: The authors use random masking to address discrepancies between the target human area and masked foreground area during inference, and the ablation proves its effectiveness. However, the random mask ratio during training could significantly impact inference results. A potential issue arises when the target human area and foreground mask during inference are entirely disjoint (e.g., the target moves left while the original foreground mask moves right). In such cases, it is unclear how the system resolves this misalignment.
2. **Foreground-Background Interaction**: When foreground actions require interaction with the background (e.g., swinging on a swing or riding a motorcycle), the system may face implicit constraints on the motion space. For instance, it might struggle to synthesize a realistic walking motion while maintaining consistency with the swing or motorcycle.

**3D Perception and Generation Limitations**

1. **Close-Up Shots**: The accuracy of tools like GVHMR and HaMeR in generating detailed 3D representations for close-ups (e.g., hands or facial expressions) may be a limitation. Even the perception models are accurate, depicting someone laughing may require precise facial blendshapes that current 3D parametric human models might not capture as well as direct video generation.
2. **Text-to-Motion’s limitations**: Text-to-motion model often generates jittery movements, which could propagate through the system and degrade the overall video quality.

**Minor points:**

- In Table 2, it is unclear how translation and orientation errors are computed. Are these errors calculated by unprojecting the values back to 3D coordinates for evaluation?
- Visually (e.g., Figure 16), some generated videos exhibit a noticeable disharmony between the foreground and background, giving the impression that the foreground is pasted onto the background.
- In Section 4.3, the authors describe temporal concatenation of reference and video tokens with RoPE modifications. This approach resembles prior methods such as OmniHuman-1, where reference injection is also handled similarly.

**Questions:**

I am particularly interested in the system's capability limits, as enhancing 3D signals improves controllability but may reduce generalization. This raises questions about the system's upper bounds and whether this explicit integration of 3D signals could become a future direction for video generation models. Additional detailed concerns are discussed in the Weakness section.

---

### Official Review · Reviewer_5sJm · 2025-11-02

**Soundness:** 3
**Presentation:** 2
**Contribution:** 3
**Rating:** 6
**Confidence:** 4

**Summary:**

This paper proposes a novel controllable video generation setup that conditions the generation on a reference image of the subject, a background video, a human trajectory on the ground, and a sequence of human poses. To the best of my knowledge, no prior work has explored this specific setup. The proposed method projects the 2D trajectory into 3D space to align with the subject’s orientation and movement speed. The resulting 3D human motion is then rendered into 2D depth, normal, and color maps, which serve as conditioning inputs for video synthesis. The visual results and supplementary videos demonstrate overall high quality, showing that the proposed method generates plausible human actions that are well integrated with the scene environment.

**Strengths:**

1. The method section presents several non-trivial and well-designed components that contribute to the effectiveness of the proposed approach. For example, Lines 195–215 describe a technique to address the foot-sliding problem. These designs are valuable, though providing more intuitive explanations would improve the readability of the paper.
2. The visual results in the supplementary videos are impressive, demonstrating high-quality human motion that is well integrated with the environment. The quantitative results in Tables 2 and 3 outperform existing baselines, and the ablation study in Table 4 clearly shows the contribution of each proposed component.

**Weaknesses:**

1. The paper adopts a relatively synthetic setup, and the required input format is not user-friendly. It needs a subject image, a background video, and a sequence of trajectory, orientation, and body poses. This means that the user-provided human motion must be naturally aligned with the background video, which would demand substantial effort from users.
2. It is unclear how users can accurately draw a 2D trajectory on the ground that aligns with the video background and can be projected into a 3D space consistent with the environment. Although the paper mentions that users can edit the trajectory and orientation according to the background, this process still appears to be very challenging in practice.
3. Given these laborious inputs, the model itself does not need to predict motion but primarily serves as a renderer. As stated in Section 2.2.4, the human mesh is used to render 2D depth, normal, and color maps that guide the video generation process. This setup closely resembles Champ [1] and MIMO [2], but these works are not included in the comparison. The authors should further clarify and highlight the differences between their method and these motion-to-video approaches.


[1] Champ: Controllable and Consistent Human Image Animation with 3D Parametric Guidance (ECCV 2024)


[2] MIMO: Controllable Character Video Synthesis with Spatial Decomposed Modeling (CVPR 2025)


4. Related works that require further discussion include:

AMG: Avatar Motion Guided Video Generation

Move-in-2D: 2D-Conditioned Human Motion Generation.

**Questions:**

1. The parentheses for all citations appear to be incorrectly formatted.
2. How many clips are included in the video dataset?
3. The paragraph from Lines 306–312 seems out of place in the overall paper structure.

---

### Note · Authors · 2025-11-14

I have read and agree with the venue's withdrawal policy on behalf of myself and my co-authors.